# Recurrent Inference Machines as Inverse Problem Solvers for MR Relaxometry

**Emanoel R. Sabidussi**[1]                                        E.RIBEIROSABIDUSSI@ERASMUSMC.NL
**Stefan Klein**[1]
**Matthan W. A. Caan**[2]
**Shabab Bazrafkan**[3]
**Arnold J. den Dekker**[3]
**Jan Sijbers**[3]
**Wiro J. Niessen**[1,4]
**Dirk H. J. Poot**[1]
[1] *Erasmus MC University Medical Center, Rotterdam, The Netherlands*
[2] *Amsterdam University Medical Center, Amsterdam, The Netherlands*
[3] *imec-Vision Lab, University of Antwerp, Antwerp, Belgium*
[4] *Delft University of Technology, Delft, The Netherlands*

**Editors:** Under Review for MIDL 2021

## Abstract

In this work, we propose the use of Recurrent Inference Machines (RIMs) to perform $T_1$ mapping. The RIM is a neural network framework that learns an iterative inference process using a model of the signal, similar to conventional statistical methods for quantitative MRI (QMRI), such as the Maximum Likelihood Estimator (MLE). Previously, RIMs were used to solve linear inverse reconstruction problems. Here, we show that they can also be used to optimize non-linear problems. The developed RIM framework is evaluated in terms of accuracy and precision and compared to an MLE method and an implementation of the ResNet. The results show that, compared to the other techniques in Monte Carlo experiments with simulated data, the RIM improves the precision of estimates without compromising in accuracy.

**Keywords:** Quantitative MRI, Deep learning, Recurrent inference machines.

## 1. Introduction

$T_1$ relaxation time is a promising biomarker for a range of diseases. Conventionally, in MR relaxometry, $T_1$ maps are estimated by fitting a known signal model to every voxel of a series of weighted images with varying contrast settings. This approach is used by widely employed methods, such as the Maximum Likelihood Estimator (MLE). However, without explicit definition of regularization priors, estimates have reduced precision if the data is noisy. In this paper, we propose a new framework for MR relaxometry based on the Recurrent Inference Machines (RIMs) (Putzky and Welling, 2017). Here, we show in Monte Carlo experiments with simulated data that the RIM estimates $T_1$ maps with higher precision than MLE and ResNet implementations.

## 2. Materials and Methods

**Recurrent Inference Machines**   The RIM is a recurrent neural network framework that learns an efficient iterative inference method and a prior that uses the neighborhood context. The framework uses the gradients of a likelihood function to plan efficient parameter updates. At a given optimization step $j \in \{0, ..., J-1\}$, the RIM receives as input the current estimate of parameters, $\hat{\boldsymbol{\kappa}}_j$, the gradient of a log-likelihood function $L$ with respect to $\boldsymbol{\kappa}$, $\nabla_{\boldsymbol{\kappa}}$, and a vector of memory states $\boldsymbol{h}_j$ to keep track of optimization progress and perform more efficient updates. The network outputs an update to the current estimate and the memory state to be used in the next iteration. The update equations for this method are given by $\{\Delta\hat{\boldsymbol{\kappa}}_{j+1}, \boldsymbol{h}_{j+1}\} = \boldsymbol{g}_\gamma(\hat{\boldsymbol{\kappa}}_j, \nabla_{\boldsymbol{\kappa}}, \boldsymbol{h}_j)$ and $\hat{\boldsymbol{\kappa}}_{j+1} = \hat{\boldsymbol{\kappa}}_j + \Delta\hat{\boldsymbol{\kappa}}_{j+1}$, where $\Delta\hat{\boldsymbol{\kappa}}_{j+1}$ is the incremental parameter update at step $j+1$ and $\boldsymbol{g}_\gamma$ represents the neural network portion of the framework, parameterized by $\gamma$. Predictions are compared to a known ground-truth and training losses are accumulated at each step. The optimal network model is learned via $\hat{\gamma} = \arg\min_\gamma (1/J) \sum_{j=0}^{J-1} \|\boldsymbol{\kappa} - \hat{\boldsymbol{\kappa}}_{j+1}\|_2^2$.

**Signal model and likelihood function**   The joint log-likelihood of the $N$ acquired images is given by $L(\boldsymbol{\kappa}, \sigma|\boldsymbol{S}) = \frac{1}{\sigma^2}\sum_{n=1}^{N}\|\boldsymbol{f}_n(\hat{\boldsymbol{\kappa}}) - \boldsymbol{S}_n\|_2^2$, where $\hat{\boldsymbol{\kappa}}$ denotes the parameter estimates, and $\sigma$ the noise standard deviation (STD). In this work, we assume $\sigma$ to be known. As signal model for the CINE sequence (Atkinson and Edelman, 1991) we use $f_n(\boldsymbol{\kappa}) = \left|A\left(1 - B\exp(-\frac{\tau_n}{T_1})\right)\right|$, in which $A$ is proportional to the proton density and receiver gain, $B$ captures the efficiency of the inversion pulse, and $T_1$ is the longitudinal relaxation time.

**Training dataset**   The RIM is trained with simulated ground truth tissue parameters $\boldsymbol{\kappa}$ and simulated weighted images $\boldsymbol{S}$. To generate training samples with a spatial distribution that resembles the human brain, ten 3D virtual brain models from BrainWeb (Aubert-Broche et al., 2006) were selected. We randomly extract 2D patches ($40 \times 40$ pixels) from the brain images, with patch centers drawn uniformly from the model's brain mask. $T_1$ values were simulated in the range of [0.3, 3.5] s and $A$ values in between [0.65, 1] a.u. $B$ values were simulated as $2 - \Gamma$, where $\Gamma$ is the half-normal distribution, with STD of 0.2. $\boldsymbol{S}$ was generated with additive zero mean Gaussian noise with STD drawn from a log-uniform distribution in the range [0.0065, 0.255] (SNR from 100 to 3). We used 31 inversion times ($\tau$) varying linearly from 0.139 to 0.937 seconds.

**Evaluation**   The RIM was compared to the MLE estimate, obtained by maximizing the $L(\boldsymbol{\kappa}, \sigma|\boldsymbol{S})$, and an implementation of the ResNet (He et al., 2015), trained with the same training data. The prediction accuracy was evaluated in terms of the Relative Bias between the reference parameter values $\boldsymbol{\kappa}$ and the estimated parameters $\hat{\boldsymbol{\kappa}}^c \in \{\hat{\boldsymbol{\kappa}}^1, ..., \hat{\boldsymbol{\kappa}}^C\}$ for each repeated experiment $c$, defined as **Relative Bias** $[\%] = \frac{1}{C}\sum_{c=1}^{C}[(\hat{\boldsymbol{\kappa}}^c - \boldsymbol{\kappa}) \oslash \boldsymbol{\kappa}] \times 100\%$, where $\oslash$ denotes the element-wise division. The Coefficient of Variation (CV) was used to measure the precision of the predictions, defined as **CV** $[\%] = \left(\text{STD}^C(\hat{\boldsymbol{\kappa}}^c) \oslash \frac{1}{C}\sum_{c=1}^{C}\hat{\boldsymbol{\kappa}}^c\right) \times 100\%$, where $\text{STD}^C$ is the standard deviation over $C$ estimates $\hat{\boldsymbol{\kappa}}$.

To assess each method's robustness to noise and mapping quality, simulated $T_1$ weighted images were generated as the training dataset, using a 2D slice of a virtual brain model not included in the training. For the same ground-truth $T_1$, $A$ and $B$ maps, $C = 100$ realisations

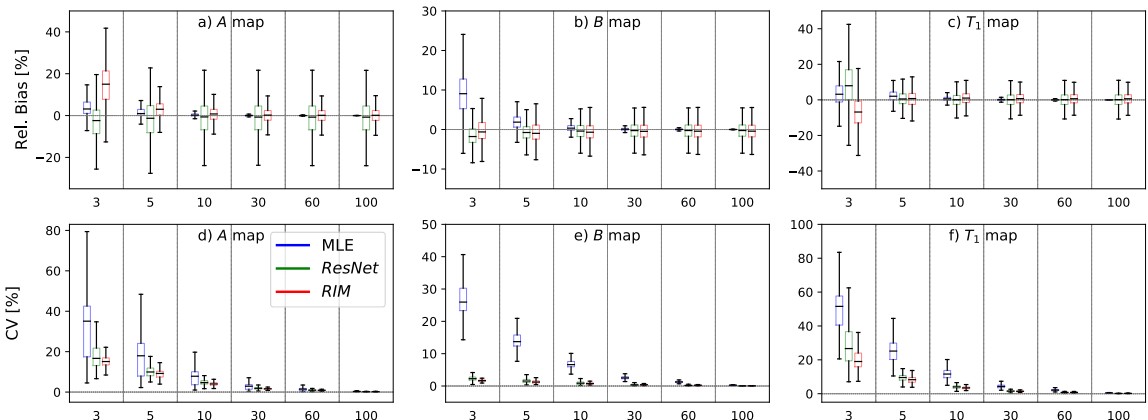

Figure 1: Results of the Monte Carlo experiment as a function of SNR levels.

of acquisition noise were simulated per SNR $\in [3, 5, 10, 30, 60, 100]$. The Relative Bias and CV were computed per pixel and their distribution over all pixels within a brain mask is presented.

## 3. Results and Conclusion

Figures 1(a)-(c) show the Relative Bias measured for $A$, $B$ and $T_1$ maps. For most cases where SNR > 3, all methods produced quantitative maps with comparable median Relative Bias, but both neural networks displayed a larger range of values than the MLE. The CV for all SNR levels is shown in Figs. 1(d)-(f) for the same data. The RIM presented lower CV than the other methods for all SNRs, while, comparatively, the MLE showed higher CV, accentuated in low SNR.

We proposed a new method for $T_1$ mapping based on the RIM framework. Experiments with simulated data show that the proposed RIM produces $T_1$ estimates with similar accuracy and higher precision than the MLE and ResNet methods. These results suggest that the RIM is a promising technique for quantitative MRI.

**Acknowledgment** This project was funded by the European Union's Horizon 2020 programme, under the grant agreement No 764513.

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
