# OpenReview forum: "Recurrent Inference Machines as Inverse Problem Solvers for MR Relaxometry"
_MIDL.io/2021/Conference/Short — MIDL 2021 Poster_

### Official Review · Reviewer_K11c · 2021-04-30

**Confidence:** 4
**Final Rating:** 3

**Summary:**

This paper proposes the use of recurrent inference machines (RIM) to estimate T1 maps. Given a signal model, the RIM uses a recurrent neural network to determine the best optimization route for updating the signal model parameters. The proposed RIM approach was compared against a traditional maximum likelihood estimation (MLE) and ResNet approach for estimating the model parameters using a simulated T1 brain images at several different noise levels and different model parameters. The RIM method resulted in higher precision of the parameter estimates.

**Strengths:**

1. The paper applies a method, RIMs, that has not seen as much use in medical image analysis applications, as far as I know.

2. The experimental evaluation involved a wide range of simulated parameter values and noise levels for thorough testing.

3. The precision results appear to be convincingly better for almost all SNR settings.

4. The paper is clearly written and easy to follow.


**Weaknesses:**

1. The relative bias of the parameter estimates, while on average appear to perform similarly to MLE, has much larger variation across different true parameter values compared to MLE. Furthermore, for the lowest SNR, the A and T1 maps have much worse bias than the MLE estimates. It is also interesting to note that unlike MLE, where increasing SNR results in reduced variation in relative bias, the larger bias of the proposed RIM method stays fairly steadily large across the range of higher SNRs.

2. There are no details regarding what the network g that performs the optimization parameter updates looks like (number of layers, size of hidden state, etc.) nor training details (when is network training stopped? convergence of parameter estimates?)



**Deanonymize Review:**

no

**Justification Of The Rating:**

While the larger range of bias is concerning, the precision/robustness of the estimates is reduced, and the applied use of RIMs for solving for physical model parameters sounds like a new and interesting idea for the medical image analysis community.

**Paper Type:**

both

**Special Issue:**

no

---

### Meta-Review · Area_Chair_gmUn · 2021-05-10

**Recommendation:** Accept (Poster)
**Confidence:** 5

**Metareview:**

The paper fits well into MIDL's short paper scope of discussing innovative ideas even if there are still some unanswered questions in the validation.

---

### Decision · Program_Chairs · 2021-05-11

Accept (Poster)